Awake fMRI reveals a specialized region in dog temporal cortex for face processing

Dilks Daniel D. 1
Cook Peter 1
Weiller Samuel K. 1
Berns Helen P. 1
Spivak Mark 2
Berns Gregory S. 1 gberns@emory.edu
1 Department of Psychology, Emory University , Atlanta, GA , USA
2 Comprehensive Pet Therapy , Atlanta, GA , USA
Vallortigara Giorgio
Electronic publication date: 2015 Aug 4
Publication date: 2015
Volume: 3
Electronic Location ID: e1115
Received 2015 May 13; Accepted 2015 Jun 30
Copyright: © 2015 Dilks et al.
Copyright year: 2015
Copyright holder: Dilks et al.
License: This is an open access article distributed under the terms of the Creative Commons Attribution License, which permits unrestricted use, distribution, reproduction and adaptation in any medium and for any purpose provided that it is properly attributed. For attribution, the original author(s), title, publication source (PeerJ) and either DOI or URL of the article must be cited.
License URL: https://creativecommons.org/licenses/by/4.0/

Keywords: fMRI, Dog, Face area

Funding: Office of Naval Research N00014-13-1-0253 This work was funded by a grant from the Office of Naval Research (N00014-13-1-0253). The funders had no role in study design, data collection and analysis, decision to publish, or preparation of the manuscript.

==============================
Recent behavioral evidence suggests that dogs, like humans and monkeys, are capable of visual face recognition. But do dogs also exhibit specialized cortical face regions similar to humans and monkeys? Using functional magnetic resonance imaging (fMRI) in six dogs trained to remain motionless during scanning without restraint or sedation, we found a region in the canine temporal lobe that responded significantly more to movies of human faces than to movies of everyday objects. Next, using a new stimulus set to investigate face selectivity in this predefined candidate dog face area, we found that this region responded similarly to images of human faces and dog faces, yet significantly more to both human and dog faces than to images of objects. Such face selectivity was not found in dog primary visual cortex. Taken together, these findings: (1) provide the first evidence for a face-selective region in the temporal cortex of dogs, which cannot be explained by simple low-level visual feature extraction; (2) reveal that neural machinery dedicated to face processing is not unique to primates; and (3) may help explain dogs’ exquisite sensitivity to human social cues.

Introduction

For social animals, faces are immensely important stimuli, carrying a wealth of information, such as identity, sex, age, emotions, and communicative intentions of other individuals (Bruce & Young, 1998; Tate et al., 2006; Leopold & Rhodes, 2010). Given the importance of face recognition for social animals, it is perhaps not surprising that humans and monkeys have dedicated neural machinery for processing visual face information discrete from the neural machinery responsible for processing nonface visual information, such as for scenes, bodies, and objects (Gross, Rocha-Miranda & Bender, 1972; Desimone et al., 1984; Perrett et al., 1988; Tsao, Moeller & Freiwald, 2008; Kanwisher & Dilks, 2013). But what about other social animals, especially non-primates, like dogs? Dogs are a special case because they are both highly social with each other and have an additional evolutionary history with humans through domestication. As such, dogs may have evolved mechanisms especially tuned to social cues and therefore may have specialized neural machinery for face processing (Hare & Tomasello, 2005; Kaminski, Schulz & Tomasello, 2012; Miklosi & Topal, 2013).

Behavioral evidence suggests that dogs may indeed process facial information (Racca et al., 2010; Somppi et al., 2014), but the neural mechanisms underlying the dogs’ behavior could be very different than humans or monkeys. For example, face recognition in dogs might rely on purely associative mechanisms, associating a face with a meaningful outcome (e.g., food). If so, then one would not expect face-specific processing in visual cortical areas, but rather activation in reward areas. Alternatively, dogs may have evolved specialized neural machinery for face recognition, and thus one would expect face-selective regions in visual cortex.

To test these competing hypotheses about face-specific processing, using fMRI, we scanned six awake, unrestrained dogs (Berns, Brooks & Spivak, 2012). To obtain high-quality fMRI data, each dog (i) completed 2–4 months of behavioral training to teach them to hold still during scanning, and (ii) had a custom-made chinrest to help minimize head movement. During scanning, dogs were presented with movie clips of human faces, objects, scenes, and scrambled objects (dynamic stimuli) and static images of human faces, dog faces, objects, scenes, and scrambled faces (static stimuli) on a projection screen placed in the rear of the magnet (Fig. 1 and Video S1).

Figure 1 Experimental setup in MRI.

Dogs were trained to station within an individually customized chin rest placed inside a stock human neck coil. The upper surface coil was located just superior to the dog’s head. Images were rear projected onto a translucent screen placed at the end of the magnet bore. In the dynamic stimuli runs, color movie clips (3-s each) were shown in 21 s blocks of human faces, objects (toys), scenes, and scrambled objects. In the static stimuli runs, black and white images (600 ms on, 400 ms off) were shown in 20 s blocks of human faces, dog faces, everyday objects, scenes, and scrambled faces. The dynamic stimuli runs were used to localize a candidate face region in the temporal cortex of dogs, and then the static stimuli runs were used to independently test the face selectivity of this region.

Materials and Methods

Participants

Participants were dogs (n = 8; 5 neutered males, 3 spayed females) from the Atlanta community. All were pets and/or released service dogs whose owners volunteer their time for fMRI training and experiments. For participation in previous experiments (Berns, Brooks & Spivak, 2012; Berns, Brooks & Spivak, 2013; Cook, Spivak & Berns, 2014), these dogs took part in a training program using behavior shaping, desensitization, habituation, and behavior chaining to prepare them to be comfortable with the physical confines of the MRI bore and the loud noise produced by scanning. Accordingly, all dogs had demonstrated an ability to remain still during training and scanning for periods of 60 s or greater.

This study was performed in strict accordance with the recommendations in the Guide for the Care and Use of Laboratory Animals of the National Institutes of Health. The study was approved by the Emory University IACUC (Protocol #DAR-2001274-120814BA), and all dogs’ owners gave written consent for participation in the study.

Training

All dogs had previously undergone training which involved the presentation of images on a computer screen (Cook, Spivak & Berns, 2014). Thus, prior to participation in the current experiment, the dogs were accustomed to viewing images on a screen in the MRI. Prior to actual scanning, all dogs underwent practice sessions with a complete run through of all stimuli (described below), which were presented in a mock scanner on a computer screen. Dogs were approved for scanning by demonstrating that they could remain motionless for the duration of at least two, 20s-blocks of black and white images of human faces, dog faces, objects, scenes, and scrambled faces, with no actual human in view. Between image blocks, dogs were either praised or rewarded with a food treat for holding still.

Experimental design

In the current experiment, a blocked fMRI design was used in which the dogs viewed either movie clips (dynamic stimuli) or static images (static stimuli). Each dog completed 3 dynamic stimuli runs and 3–4 static stimuli runs, depending on their tolerance of the MRI. In the dynamic runs, dogs were presented with 3-s color movie clips of human faces, objects (toys), scenes, and scrambled objects (Pitcher et al., 2011) (Fig. 1 and Video S1). The scrambled object movies were constructed by dividing each object movie clip into a 15 by 15 box grid and spatially rearranging the location of each of the resulting movie frames. There were 7 movie clips for each category. Each run contained two sets of four consecutive stimulus blocks in palindromic order (e.g., faces, objects, scenes, scrambled objects, scrambled objects, scenes, objects, faces), to make two blocks per stimulus category per run. In the static runs, dogs were presented with black and white images of human faces, dog faces, objects, scenes, and scrambled faces. The scrambled face images were constructed using the steerable pyramid method (Simoncelli & Freeman, 1995). Each image was presented for 600 ms followed by a 400 ms black screen interstimulus interval. There were 20 images for each category. Each run contained two sets of five consecutive stimulus blocks in palindromic order (e.g., human faces, dog faces, objects, scenes, scrambled faces, scrambled faces, scenes, objects, dog faces, human faces), to make two blocks per stimulus category per run.

MRI scanning

All scanning was conducted with a Siemens 3 T Trio whole-body scanner. Head movement was reduced by a custom chinrest for each dog that allowed the dog to achieve consistent stationing in the MRI coil (a standard neck coil) (Berns, Brooks & Spivak, 2013). All participants wore ear plugs during scanning. Each scan session began with a 3s, single image localizer in the sagittal plane (SPGR sequence, slice thickness = 4 mm, TR = 9.2 ms, TE = 4.15 ms, flip angle = 40 degrees, 256 × 256 matrix, FOV = 220 mm). A T2-weighted structural image was previously acquired during one of our earlier experiments using a turbo spin-echo sequence (25–30 slices, TR = 3,940 ms, TE = 8.9 ms, voxel size = 1.5 × 1.5 × 2 mm, flip angle = 131 degrees, 26 echo trains), which lasted ∼30 s.

Functional scans used a single-shot echo-planar imaging (EPI) sequence (24 slices, TR = 1,400 ms, TE = 28 ms, voxel size, 3 × 3 × 3 mm, flip angle = 70 degrees, 10% gap). Slices were oriented dorsally to the dog’s brain (coronal to the magnet, as, in the sphinx position, the dogs’ heads were positioned 90 degrees from the usual human orientation) (Fig. 1) with the phase-encoding direction right-to-left. Sequential slices were used to minimize between-plane offsets from participant movement, and the 10% slice gap minimized the crosstalk that can occur with sequential scan sequences. We have previously found that both structural and functional resolutions were adequate for localizing activations to structures like the caudate nucleus (Berns, Brooks & Spivak, 2012; Berns, Brooks & Spivak, 2013; Cook, Spivak & Berns, 2014).

Stimuli were presented using Python 2.7.9 and the Expyriment library. Each stimulus block was manually triggered by an observer at the rear of the magnet. This manual triggering ensured that the dog was properly stationed at the beginning of each block. Importantly, no actual human was in view during any of the stimulus presentation blocks. The center of each stimulus was presented binocularly, and at eye level in front of the dog, such that each stimulus fell in the center of the visual field when the dog was looking forward.

Heart rate and respiration rate were not collected during scanning. However, we do not believe that either of these physiological measures contributed to our findings for two reasons. First, although dogs’ heart rate (HR) and respiration rate (RR) are greater than humans’, they are not that much faster. In fact, large dogs (which most of our subjects are) have HRs and RRs similar to humans. Additionally, the TR in this study was 1,400 ms, which is about 30% faster than the TR in a typical human study (2,000 ms). Thus, any modestly faster HR and RR is compensated for by the faster TR. Second, both HR and RR would produce a general effect across brain regions, and we see differential effects between DFA and V1 (see Functional Data Preprocessing and Analysis Section and Results Section).

Functional data preprocessing and analysis

Preprocessing was conducted using AFNI (NIH) and its associated functions, and steps were identical to those described previously (Berns, Brooks & Spivak, 2012; Berns, Brooks & Spivak, 2013). In brief, 2-pass, 6-parameter affine motion correction was used with a hand-selected reference volume for each dog. We hand selected a reference volume because the first volumes are never the most representative of the dog’s head position during the study. The reference volume was typically midway in the first run, after the dog had settled into a comfortable position. Next, because dogs moved between blocks (and when rewarded), aggressive censoring was carried out. A volume was flagged for censoring based on two criteria: (1) excessive scan-to-scan motion; and (2) identification as an outlier based on overall signal intensity. Censored files were inspected visually to be certain that bad volumes (e.g., when the dog’s head was out of the scanner) were not included. The majority of censored volumes followed the consumption of food. If less than 33% of the volumes were retained, we excluded that subject (Berns, Brooks & Spivak, 2013). This resulted in the exclusion of two dogs. For the remaining six dogs, 47–76% of volumes were retained.

EPI images were then smoothed and normalized to %-signal change. Smoothing was applied using 3dmerge, with a 6 mm kernel at Full-Width Half-Maximum (FWHM). The resulting images were then input into the General Linear Model. For each subject, a General Linear Model was estimated for each voxel using 3dDeconvolve. For the dynamic stimuli runs, the task-related regressors in this model were: (1) human faces, (2) objects, (3) scenes, (4) scrambled objects, and 5) fixation immediately prior and after each stimulus block. For the static stimuli runs, the task-related regressors in this model were: (1) human faces, (2) dog faces, (3) objects, (4) scenes, (5) scrambled faces, and 6) fixation immediately prior and after each stimulus block. Because our previous work measuring the hemodynamic response function (hrf) in dogs on this task revealed a peak response at 4–6 s after signal onset (Berns, Brooks & Spivak, 2012), the six task regressors were modeled as variable duration events and were convolved with a single gamma function approximating the hrf. Motion regressors generated by the motion correction were also included in the model to further control for motion effects. A constant and linear drift term was included for each run.

To identify a candidate dog face area (DFA) for each dog, we used the dynamic stimuli runs (contrast: faces > objects). Next, we identified the voxel with maximal intensity in the temporal lobe ventral and caudal to the splenium (Datta et al., 2012). We then placed a spherical ROI of 5 mm radius around the this voxel for subsequent testing of the static stimuli. For V1, we identified the area of maximal activation in the dorsal occipital region when comparing the mean signal of all categories in the dynamic condition to baseline and placed an ROI of 5 mm radius around this voxel. ROI locations were placed by one author and confirmed by two others. The mean signal was then extracted from these ROIs for each dog in each of the 5 conditions of static stimuli. We used a mixed-effect model (SPSS 21, IBM) to determine whether there was a significant effect of stimulus category in both the putative DFA and V1 (maximum-likelihood estimation, scaled identity covariance structure for repeated effects and subject for random effect).

The average timeseries during stimulus blocks for faces and objects was extracted from the DFA for each dog. Each timeseries was detrended, and values from censored volumes replaced with NaNs. The activity within the ROI immediately preceding the onset of the stimulus block was subtracted from subsequent values during the stimulus presentation. This compensated for any residual activity during the interstimulus intervals. The resultant timeseries were averaged over all repetitions and dogs.

Results

The DFA was detected in the right hemisphere in all six dogs, but differed slightly in the medial-lateral direction: Four dogs exhibited a DFA more medially, while the other two dogs exhibited a DFA more laterally. Next, a region-of-interest (ROI) of 5 mm radius was centered over the peak voxel within the predetermined DFA for each dog (Fig. 2), and the activity of this ROI was compared across the stimulus categories in the static stimuli runs. Crucially, data from the static stimuli runs served as the test data and were independent from the data used to define the DFA. First, we found no significant difference in activation between human and dog faces in the DFA (t(5) = 0.82, p = 0.44), and thus collapsed across these two categories. Second, to investigate face selectivity, we then compared the response to images of faces to images of objects, scenes, and scrambled faces in DFA, and found a significant category effect (F(3, 24) = 3.79, p = 0.02), with a significantly greater response to images of faces compared to objects (punc = 0.004; pBonf = 0.01), a marginally greater response than scenes (punc = 0.06; pBonf = 0.17), but no significant difference in response to scrambled faces (punc = 0.30; pBonf = 0.89) (Fig. 3A). Extraction of average timecourses from DFA to both faces and objects showed that objects decayed quickly while the response to faces was sustained, resulting in an overall greater response (Fig. 4). These findings not only reveal within-subject replicability across paradigms, but also the face selectivity of the DFA, namely its significantly greater response to faces compared to objects.

Figure 2 ROI locations for the dog face area (DFA) and primary visual cortex (V1).

The DFA was identified by the contrast of faces versus objects during the dynamic stimuli runs. Each color represents the ROI of one dog. For visualization and comparison of location, the ROIs have been spatially normalized and overlaid on a high resolution dog brain atlas (Datta et al., 2012). The location of the DFA was localized to the medial bank of the ventrocaudal temporal lobe in 4 of the 6 dogs, with the other 2 localized more laterally. V1 was identified by the average of all dynamic run conditions (face, objects, scenes, scrambled) relative to baseline. In each dog, a dorsal area of activation in the caudal portions of the marginal/endomarginal gyri was identified and corresponded to the known location of primary visual cortex.

Given the similarity in low-level features between faces and scrambled faces, it is not surprising that the DFA might respond to scrambled faces, albeit less reliably than to the images of faces themselves. But might the face selectivity in the DFA be explained entirely by retinotopic information simply inherited from early visual cortex? To address this possibility, we defined the primary visual cortex (V1) (contrast: average of all stimulus categories versus baseline) using the dynamic stimuli runs for each dog. For all subjects, we found a region dorsally in the caudal portion of the marginal and endomarginal gyri, consistent with the known location of dog V1 (Beitz & Fletcher, 1993). Next, an ROI of 5 mm radius was centered over the peak voxel within the predetermined V1 for each dog, and the activity of this ROI was compared across the stimulus categories in the static stimuli runs. The face selectivity of this V1 ROI was then compared to the face selectivity of the DFA. A 2 (ROI: DFA, V1) × 2 (Condition: faces, objects) mixed-effect model revealed a significant interaction (F(1, 30) = 6.68, p = 0.02), indicating that the face selectivity of the DFA was not like that of V1, and thus not strictly a result of low-level feature extraction (Fig. 3B).

Figure 3 Average percent signal change in DFA and V1.

Error bars indicate the standard error of the mean (n = 6). (A) In DFA, we found a significant category effect (F(3, 24) = 3.79, p = 0.02), with a significantly greater response to images of faces compared to objects (∗∗ p = 0.004) and a marginally greater response to scenes (∗p = 0.06). (B) V1 had a similar level of response to all stimulus categories (F(3, 24) = 0.42, p = 0.74), and crucially was significantly different from DFA in face selectivity (i.e., faces compared to objects) (F(1, 30) = 6.68, p = 0.02).

But might stimulus-correlated motion (e.g., the dogs moved more on images of faces than objects) explain our results? Stimulus-correlated motion would produce a general effect across brain regions, and we, in fact, see a differential effect between DFA and V1, thus we do not believe that stimulus-correlated motion provides an alternative account. However, to confirm that stimulus-correlated motion did not bias our results, we calculated the mean scan-to-scan motion as the Euclidean sum of differential translations in the three principal directions. The overall mean scan-to-scan motion was 0.62 mm (s.e. = 0.09), but this measure was not significantly different across the 4 stimulus categories (F(3, 24) = 0.43, p = 0.73). Thus, dogs did not move more during one category versus another, and as such stimulus-correlated motion cannot explain our results. (The same result was obtained for the censored trials (mean = 0.22 mm, s.e. = 0.03) (F(3, 24) = 0.27, p = 0.84)).

Discussion

Taken together, the above results provide the first evidence for a region in temporal cortex of dogs involved in the processing of faces. Indeed, while there is ample behavioral evidence that dogs respond to faces, our results demonstrate an evolutionary continuity in the neural substrates of a key aspect of social behavior: a face-selective region in dog cortex located in an area similar to that of humans and monkeys. The commonality of location is consistent with the commonality and importance of face processing in social species and is found in visual cortex, suggesting that dogs’ ability to process faces is not simply the result of learned associations. Our finding that dogs, like humans and monkeys, exhibit specialized cortical face regions is also consistent with two other studies demonstrating that neural machinery dedicated to face processing may not be unique to primates, having been observed in sheep (Kendrick & Baldwin, 1987) and crows (Marzluff et al., 2012).

In addition to behavioral evidence suggesting dedicated neural machinery for face processing in dogs, one previous study suggested a neural signature for such processing (Tornqvist et al., 2013). Using visual event-related potentials (ERPs), this study reported differences in two ERP components between the responses to human and dog faces. This study gave the first hint of a neural substrate for face processing but also raised several questions, namely the degree to which visual recognition of any object (given only face stimuli were tested) or low-level feature extraction (given the low-level visual differences between the images of human and dog faces) might explain the results. Moreover, the limited spatial resolution of EEG precluded the precise localization of putative face-selective machinery, which is relatively small and restricted to specific regions of occipital and temporal cortex in primates. Our fMRI results build on these ERP findings and offer strong evidence for a face-selective region in dog temporal cortex, responding significantly more to images of faces than to images of objects. Furthermore, the face selectivity of the DFA was not found in dog primary visual cortex, ruling out simple low-level feature extraction as explanations for the face-selective response in DFA.

The principal limitation of our study stems from the small effect size. The average differential BOLD response was well less than 1%, which is consistent with human fMRI studies. Comparable animal fMRI studies, however, overcome the signal limitation by immobilizing the subject and scanning for much longer periods of time to decrease the effects of noise, but this approach often uses a small number of subjects – typically two. In contrast, our approach is to use awake, trained dogs who cooperatively enter the scanner and hold still for periods up to several minutes without restraint. And while the dogs do extraordinarily well, the data quality cannot approach that obtained from a sedated, immobilized monkey. Thus, the trade-off is noisier data. We compensate by using more subjects than a typical monkey study, here reporting the data from six dogs. Although we have studied larger cohorts of dogs in previous studies, watching images on a flat screen is not a natural behavior for dogs, and only a subset of the MRI-trained dogs would do so, even after months of training. Even so, the data we report here show a high degree of within-subject replicability, with some inter-subject variation in the location of DFA, some of which may be due to noise and some due to the existence of multiple face-sensitive patches. Another potential limitation of our study may be the concern about vasculature effects. In fMRI, signal changes in a given region of cortex are attributed to neuronal activity. However, it could be the case that such fMRI signal changes might arise from a draining vein, making it difficult to say whether the fMRI signal changes are due to neuronal activity in that region of cortex, in more distant cortical regions, or both. Physiological noise may hypothetically affect the detected activations; however, we have no a priori reason to suspect that the reported DFA or V1 activities are due to physiological confounds. The lack of condition-specific effects in V1 rules out a global confound, so the remaining question is whether the putative DFA activity is a result of physiologic noise on a local level. In a previous dog-fMRI study (Cook, Spivak & Berns, 2014), we investigated the inclusion of a ventricle ROI as a covariate and proxy for physiological noise and found that it was not a significant contributor to activations in the reward system. In that experiment, the stimuli represented conditioned signals to food reward, and would be expected to be far more arousing than the visual stimuli used here. Thus, the ultimate goal is to obtain converging evidence across multiple methodologies (fMRI, neurophysiology, lesion studies, etc.) and across multiple labs to definitively establish the selectivity of a given cortical region.

Finally, it is important to realize that a “baseline” does not exist in this study—at least in the same way as in human and monkey studies. Because the dogs were unsedated and unrestrained, they were periodically reinforced with treats and praise for holding still, and because this occurred between stimulus blocks, one should not interpret the baseline as “nothing”. In fact, the dogs were always on task, whether a stimulus was visible or not. Although we report BOLD activations relative to an implicit baseline (Figs. 3 and 4), the statistics and inferences are based on the differential activity between stimulus conditions (e.g., faces and objects).

Figure 4 Average time course of activation in DFA for faces and objects.

The stimulus was visible for 20 s. Each time course was referenced to the volume immediately preceding the onset of the stimulus and was averaged over all dogs and all trials (excluding censored volumes). The response to objects decayed quickly while the response to faces was sustained, resulting in an overall greater response, which was individually significant at the indicated time points (∗ t > 1.65).

In summary, the existence of a face-selective region in temporal dog cortex opens up a whole range of new questions to be answered about their social intelligence: What is the relative impact of early socialization on dog versus human face processing? Do face regions in dogs process emotional content? Given canids’ reliance on body posture for communication, are there corresponding body-selective regions? We do not know whether face-selective cortex in dogs is a result of the domestication process and dogs’ resultant reliance on humans, or whether such face regions predate domestication and exist widely in other social carnivores. But the relatively small size of the dog brain, and the dedication of face processing to specific regions, highlights the importance of face processing to this species, and may explain dogs’ exemplary skill at interspecies communication.

Supplemental Information

Video S1 Sample videos of dynamic localizer

Click here for additional data file.

Supplemental Information 2 Extracted ROI values in DFA and V1 for all dogs

Click here for additional data file.

Supplemental Information 3 Sample consent form for participation in the study

Click here for additional data file.

We are grateful to the dogs’ owners for the time they have devoted to training: Cindy Keen (Jack), Patricia King (Kady), Nicole Zitron (Stella), Darlene Coyne (Zen), Marianne Feraro (Eddie), and Cory and Anna Inman (Tallulah).

Additional Information and Declarations

Competing Interests

Author Contributions

Animal Ethics

Data Availability

Mark Spivak is president of Comprehensive Pet Therapy. Gregory Berns and Mark Spivak own equity in Dog Star Technologies and developed technology used in the research described in this paper. The terms of this arrangement have been reviewed and approved by Emory University in accordance with its conflict of interest policies.

Daniel D. Dilks conceived and designed the experiments, analyzed the data, wrote the paper, prepared figures and/or tables, reviewed drafts of the paper.

Peter Cook conceived and designed the experiments, performed the experiments, analyzed the data, contributed reagents/materials/analysis tools, prepared figures and/or tables, reviewed drafts of the paper.

Samuel K. Weiller and Mark Spivak conceived and designed the experiments, performed the experiments, reviewed drafts of the paper.

Helen P. Berns conceived and designed the experiments, contributed reagents/materials/analysis tools, reviewed drafts of the paper.

Gregory S. Berns conceived and designed the experiments, performed the experiments, analyzed the data, wrote the paper, prepared figures and/or tables, reviewed drafts of the paper.

The following information was supplied relating to ethical approvals (i.e., approving body and any reference numbers):

This study was performed in strict accordance with the recommendations in the Guide for the Care and Use of Laboratory Animals of the National Institutes of Health. The study was approved by the Emory University IACUC (Protocol #DAR-2001274-120814BA), and all dogs’ owners gave written consent for participation in the study.

The following information was supplied regarding the deposition of related data:

http://dx.doi.org/10.5061/dryad.8qv09.

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
