# Peer review of "Awake fMRI reveals a specialized region in dog temporal cortex for face processing"

_PeerJ, doi:10.7717/peerj.1115_

## Round 0.1 · original submission · Major Revisions

Both Reviewers stressed a number of methodological problems that need to be properly addressed before any final decision about the suitability of your paper to be published can be reached.

Reviewer 1 ·

Basic reporting

No comments.

Experimental design

Concerning the blocked fMRI design: a blocked design is optimized for sensitivity, however, at the cost of specificity. Both, stimulus correlated motion and stimulus correlated BOLD signal fluctuations in medium and large draining veins, represent serious confounds and cannot be corrected using the author's design and data preprocessing. So I would ask the authors to comment on this and to justify their choice of the blocked fMRI design.

Validity of the findings

I see major problems for the validity of the findings, which I list in the following:
.) MRI scanning: Head movement was probably reduced but not minimized (no comparison or else provided).
.) The spatial resolution of structural images is rather poor as compared to humans, taking into account the much smaller size of dog brains (likely due to frequent head movements and, subsequently, minimized scan time).
.) Functional scans also show poor spatial resolution and long TR, taking into account the dog's higher heart rate. Therefore, the authors must expect a stronger contribution from physiological noise, further reducing sensitivity and specificity. Please comment.
.) Functional data preprocessing: hand-selected definition of a reference volume and (agressive) censoring represents a serious bias in the experiment, please comment.
.) Smoothing of EPI images, although rather common, should be avoided as this adds to a further loss of specificity.
All these limitations need to be addressed in the Discussion section, including suggestions for improvement.

Additional comments

To summarize the methodological limitations and taking into account the approximately 50% of data not eliminated by the authors themselves, the remaining low quality data sets, as compared to established data quality in humans, do not allow any reasonable scientific interpretation. The data may be considered as pilot data only.

·

Basic reporting

The manuscript lacks an indication of the database where raw data will be made available. I think it is crucial that raw data from fMRI scans are made available to the scientific community such as to make sure other investigators can reproduce the authors' findings.
I have otherwise no comments

Experimental design

The work builds heavily on previous research form the authors.
How did the authors identify areas of "peak activation"? What statistical threshold did they use and on what statistical ground? Did they use a fixed or mixed effect model? I think these important parameters should be reported. Also, no mention of the strategy employed to correct images for familywise errors is reported.

I would suggest to include BOLD signal timecourses from affects areas and non affected regions to inform readers also on magnitude as well as temporal dynamics of the effects.

Validity of the findings

In the light of the crucial experimental omissions reported above, it is not easy to comment on the statistical robustness of the identified effects. I think more detail on the statistical analyses employed should be included in a revised versions.
No discipline-specific repository where the data can be made available is reported.

---

## Round 0.2 · Minor Revisions

Reviewer 2 is happy with your revision, whereas Reviewer 1 asked for some further minor amendments.

Reviewer 1 ·

Basic reporting

No comments.

Experimental design

No comments.

Validity of the findings

No comments.

Additional comments

Here I comment on the responses of the authors ("Response to reviews") to my comments and criticism of the original manuscript.

The authors are correct that various physiological signal contributions (eg, venous drainage, heart beat, respiration, involuntary head motion) may contribute to non-localized signal changes. However, they are mislead in case they assume that these signals would be smeared across the brain evenly and thus "only" reduce overall functional contrast to noise (which is bad enough). Due to the nature of these contributions they may be (much) stronger in one location than in another which may then affect "activations" in one region (eg, DAF) differently than in the other (eg, visual cortex). Therefore, the authors should be more attentive towards potential artifacts (examples and discussions may be found in the 20+ year human fmri literature by Triantafyllou et al. 2005, 2011 a,b, Gati et al. 1997 ff, Tong et al. and many more).

Concerning the author's approach using manually predifined ROIs, which was or is not so uncommon in fMRI, I have two comments:
a) usually, this is done in case a whole-brain group analysis (FWE or FDR corrected) does not provide useful results (the author's argument that dog brains are too different is not a strong one when used to work with humans), and
b) if this ROI approach is used at all, there should be a consensus analysis (very common in radiology) performed where 3 independent researcher selected the ROIs and the outcome should be analyzed in terms of variability and robustness.

I hope these comments are helpful to further improve the quality of the paper.

·

Basic reporting

No Comments

Experimental design

No Comments

Validity of the findings

No Comments

---

## Round 0.3 · accepted · Accept

I believe you addressed successfully all the reviewers comments and that the paper is now acceptable for publication.